# Microstructure and Oxygen Evolution Property of Prussian Blue Analogs Prepared by Mechanical Grinding

**DOI:** 10.3390/nano13172459

**Published:** 2023-08-30

**Authors:** Abhishek Meena, Chinna Bathula, Mohammad Rafe Hatshan, Ramasubba Reddy Palem, Atanu Jana

**Affiliations:** 1Division of Physics and Semiconductor Science, Dongguk University-Seoul, Seoul 04620, Republic of Korea; pakar.abhishek@gmail.com; 2Division of Electronics and Electrical Engineering, Dongguk University-Seoul, Seoul 04620, Republic of Korea; cdbathula@dongguk.edu; 3Department of Chemistry, College of Science, King Saud University, P.O. Box 2455, Riyadh 11451, Saudi Arabia; mhatshan@ksu.edu.sa; 4Department of Medical Biotechnology, Dongguk University, Goyang 10326, Republic of Korea; palemsubbareddy@gmail.com

**Keywords:** Prussian blue, mechanochemical grinding, OER, stability

## Abstract

Solvent-free mechanochemical synthesis of efficient and low-cost double perovskite (DP), like a cage of Prussian blue (PB) and PB analogs (PBAs), is a promising approach for different applications such as chemical sensing, energy storage, and conversion. Although the solvent-free mechanochemical grinding approach has been extensively used to create halide-based perovskites, no such reports have been made for cyanide-based double perovskites. Herein, an innovative solvent-free mechanochemical synthetic strategy is demonstrated for synthesizing Fe_4_[Fe(CN)_6_]_3_, Co_3_[Fe(CN)_6_]_2_, and Ni_2_[Fe(CN)_6_], where defect sites such as carbon–nitrogen vacancies are inherently introduced during the synthesis. Among all the synthesized PB analogs, the Ni analog manifests a considerable electrocatalytic oxygen evolution reaction (OER) with a low overpotential of 288 mV to obtain the current benchmark density of 20 mA cm^−2^. We hypothesize that incorporating defects, such as carbon–nitrogen vacancies, and synergistic effects contribute to high catalytic activity. Our findings pave the way for an easy and inexpensive large-scale production of earth-abundant non-toxic electrocatalysts with vacancy-mediated defects for oxygen evolution reaction.

## 1. Introduction

Halide-based perovskites have gained momentum for energy applications such as solar cells and light-emitting diodes [1,2,3]. However, their instability in water restricts them from being considered as electrocatalysts, whereas oxide and cyanide-based perovskites are considered potential candidates for oxygen and hydrogen evolution reactions [4,5]. Cyanide-based DP compounds such as PB and PBAs are essential materials that can meet the large-scale energy demand, mainly from fossil fuels. Although noble-metal-based catalysts, such as IrO_2_ [6] and RuO_2_ [7], show efficient electrocatalytic performance for oxygen evolution reaction (OER), their scarcity and high cost have been considered a barrier to large-scale application. Therefore, it is appealing but challenging to design and synthesize earth-abundant and low-cost electrocatalysts for OER [8,9,10,11]. The development of electrocatalysts for water splitting using inexpensive and abundant elements is of utmost importance for advancing sustainable energy technologies. Water splitting, particularly through electrolysis, is a key process in producing hydrogen gas, which can be used as a clean and versatile energy carrier, especially when generated using renewable sources like solar or wind power. However, the cost and scarcity of catalyst materials have been significant barriers to large-scale adoption of water-splitting technologies [12,13,14].

Prussian blue, Fe_4_[Fe(CN)_6_]_3_·nH_2_O, is a mixed-valence compound that contains both ferrous (Fe^2+^) and ferric iron (Fe^3+^) [15]. Cyanide ions serve as a bridge between the iron ions, which are arranged alternately in a face-centered cubic lattice. Each Fe^3+^ and Fe^2+^ ion has octahedral geometry; Fe^3+^ and Fe^2+^ ions are surrounded by nitrogen and carbon of cyanide ions, respectively. Due to the 3:4 ratio of Fe^2+^ and Fe^3+^ sites in the lattice, there is a 25% vacancy of [Fe^2+^(CN)_6_]^4−^ clusters. There is a vacant 12-fold cuboctahedral position where alkali metals can be inserted, creating double perovskite like the structure of Prussian blue analogs (PBAs). The generic formula of Prussian blue analogs (PBAs) is represented as A_p_M^(a)^[M^(b)^(CN)_6_]_y_·mH_2_O, (M^(a)^, M^(b)^ = Fe, Mn, Co, Ni, Cu, Zn, and A = Li, Na, K, Rb) where M^(a)^ and M^(b)^ are the metal ions coordinated to cyano groups and A cations are intercalated between the frameworks. Prussian blue was synthesized through gas-phase synthesis, solution-phase synthesis, and template synthesis [16,17]. However, the preceding methods entail the following problems: long synthetic time, solvents, other chemicals (such as surfactants or templates), complex process control and steps, or high reaction temperature or pressure. By contrast, solid-phase synthesis can be completed in one step at room temperature without using solvents. By modifying both cation and anion defect sites, it is possible to create a variety of fascinating functions in materials, including band structure, magnetism, conductivity, and catalysis.

Defects have gained increasing attention from researchers in recent years for their substantial impact on oxygen electrocatalysis—a vital process in applications like fuel cells, metal–air batteries, and oxygen reduction reactions in electrochemical devices [18,19,20]. These defects, prevalent in materials like carbon-based catalysts and transition metal oxides, boost catalytic activity by creating active sites that expedite oxygen species adsorption and reduction. Furthermore, defects influence selectivity by favoring specific reaction pathways, enhance durability by stabilizing catalysts and prolonging device lifespan, and are induced through heteroatom doping, introducing unique electronic properties that augment catalytic activity. Investigating these defects provides valuable insights into underlying reaction mechanisms and kinetics, enabling tailored catalyst design at the nanoscale to optimize oxygen electrocatalysis. Ultimately, harnessing defect potential holds promise for advancing clean energy technologies, improving energy conversion and storage device efficiency. Defect-mediated vacancies of oxygen, sulfur, iodine, nickel, and iron are well-known for various applications. PB and PBAs have shown promising applications in electrocatalytic water-splitting reactions due to their unique structural features, such as nanoporous open framework structures, large specific surface areas, and low-cost and easy preparation. These PBAs are also used as a cathode and anode for fabricating electrolyzers for overall water splitting [21,22] and batteries [23,24,25,26,27]. The surface area of catalysts is closely tied to their electrochemical activity, particularly in applications like fuel cells, batteries, and electrocatalytic reactions. A large surface area is crucial because it provides more active sites for electrochemical reactions to occur, thereby increasing electrochemical activity. A larger surface area also enhances the accessibility of reactant molecules to these active sites, facilitating adsorption and participation in reactions. In practical applications, a larger surface area allows for greater catalyst loading onto support materials, improving overall catalytic activity. Efficiency benefits from a larger surface area, reducing the need for costly catalyst materials. Additionally, it influences reaction kinetics, leading to faster electrochemical processes, but must be balanced with considerations of catalyst durability. Altogether, optimizing surface area and active site distribution is fundamental for designing efficient catalysts in various electrochemical contexts, including energy conversion and storage systems [28,29,30,31].

Herein, we report the easy and cost-effective mechanochemical synthesis of Fe_4_[Fe(CN)_6_]_3_ (S1), Co_3_[Fe(CN)_6_]_2_ (S2), and Ni_2_[Fe(CN)_6_] (S3) (Figure 1 and Appendix A). The low formation energy of metal chloride salts and metal cyanide ions facilitates the formation of PB and various PBAs. It should be mentioned that halide-based perovskites are unstable in water. However, our synthesized cyanide-bridged DPs are highly stable in water. This allows us to use them as an OER catalyst. We have shown the potential applications of the synthesized DPs as an electrocatalyst. For this purpose, we have chosen three compounds S1, S2, and S3. We found that S3 showed the best OER catalytic activity. Synergistic effects and the CN vacancies have been inherently introduced during the synthesis of these materials, and these defects are responsible for their excellent catalytic activity.

## 2. Materials and Methods

### 2.1. Materials

Alfa-Aesar was the source for purchasing anhydrous, 99.5% metal basis iron chloride(II) (FeCl_2_), ultra-dry, 99.998% metal basis cobalt chloride (CoCl_2_), and anhydrous, 99.99% metal basis nickel chloride (NiCl_2_). Meanwhile, Sigma-Aldrich, St. Louis, MO, USA, was the supplier for reagent grade iron chloride(III) (FeCl_3_), and powder, 99% potassium ferrocyanide (K_4_[Fe(CN)_6_]) and potassium ferricyanide (K_3_[Fe(CN)_6_].

### 2.2. Synthesis of PB and PBAs

First, 100 mg of FeCl_3_ and 195 mg of K_4_[Fe(CN)_6_] were added to a mortar and ground with a pestle. The mixture immediately changed color and was ground for 10 min until it was evenly colored. Residuals were washed out with water and the resulting moist powder was filtered using an air pump. The powder was then transferred to a vacuum oven and dried for 8 h at 100 °C. S2 and S3 were prepared following the above procedure.

## 3. Results and Discussion

This study presents a novel, one-step solid-state synthesis method for preparing Fe-based double perovskite PB and PBAs. In this method, we ground the precursor salts for 10 min, resulting in an immediate color change indicating a solid-state reaction. We then washed the product thoroughly with water and dried it under a vacuum at 100 °C. Notably, we observed significant color changes attributed to the encapsulation of crystallization water within the double perovskite cages.

We further investigated the optical properties of the synthesized compounds through absorption spectroscopy (Figure 2a). Our results show that S1 exhibited an absorption maximum at 641 nm, while S2 and S3 displayed absorption maxima at 525 nm and 641 nm, respectively. These distinct absorption peaks are likely due to the intramolecular charge transfer phenomena through cyanide bridging, which is known to significantly alter the energy of charge transfer events. Additionally, the observed variation in absorption peaks may be attributed to differences in the ionic potential of Fe^2+^, Co^2+^, and Ni^2+^ in the various compounds.

We conducted FTIR analysis on Prussian blue to determine the presence of cyanide and water molecules, as shown in Figure 2b. Our analysis revealed a strong FTIR peak at 2072 cm^−1^, corresponding to the cyanide group between iron cations. We also observed peaks at 1604 cm^−1^ and 1403 cm^−1^, indicating the presence of interstitial water molecules [17]. To assess the thermal stability of the synthesized compounds, we carried out TGA, which is presented in Figure 2c. Our results show that all compounds underwent the first decomposition around 100 °C, which is attributed to the loss of crystallization water [32]. Furthermore, S1, S2, and S3 exhibited the second decomposition at around 220 °C, 330 °C, and 525 °C, respectively. These findings suggest that these compounds are highly stable and can be suitable for various catalytic applications.

To confirm the complete conversion of PBA from the respective precursor salts, we performed PXRD, XPS, and SEM. The PXRD patterns of S1, S2, and S3 are depicted in Figure 3a–c. Notably, we did not observe any peaks corresponding to the precursors (FeCl_3_ and K_4_[Fe(CN)_6_]) used in the synthesis of Prussian blue in the resulting product (Figure 3a–c).

The PXRD analysis of the as-prepared Prussian blue (S1) showed characteristic peaks at 17.4° (200), 24.7° (220), 35.3° (400), and 39.4° (420), which are indicative of the face-centered cubic (fcc) phase of Fe_4_[Fe(CN)_6_]_3_. These results were consistent with the standard patterns of Prussian blue crystals (Fe_4_[Fe(CN)_6_]_3_, JCPDS card 73-0687), confirming the successful conversion of the precursors into double perovskite compounds. We further examined the shape, size, and distribution of Prussian blue analog particles using high-resolution scanning electron micrographs (FESEMs) of S1, S2, and S3 nanoparticles, as shown in Figure 3d,e. Our images at a 500 nm scale in Figure 3d,e confirm that all the double perovskites are aggregates of nanoparticles with no specific shape or size.

To investigate the element valences and surface chemical composition of the compounds, X-ray photoelectron spectroscopy (XPS) measurements were carried out and the XPS plots are shown in Figure 4a–f. Our results in Figure 4a demonstrate the existence of Fe, Ni, C, and O without any impurity elements, indicating the formation of S3. The Fe 2p XPS spectrum exhibits two peaks located at 707.86 and 720.69 eV, that are assigned to Fe 2p_3/2_ and Fe 2p_1/2_, respectively, indicating the presence of Fe^3+^ [33] (Figure 4b). 

The Ni 2p XPS spectrum in S3 demonstrates two XPS peaks of Ni 2p_3/2_ and Ni 2p_1/2_ at 855.88 and 873.67 eV, respectively, with satellite peaks of Ni^2+^ at 861.18 and 879.58 eV (Figure 4c) [34]. The high-resolution C 1s XPS peaks at 283.88 eV and 284.68 (Figure 4d), and the main peak of the N 1s in Figure 4e at 396.88 and 397.58 eV, indicate the presence of C–N in S3. In S1, the Fe 2p peaks at 708.48 eV and 721.38 eV are of Fe^2+^_3/2_ and Fe^2+^_1/2_, respectively, while the Fe 2p peaks at 712.58 eV and 725.08 eV are signatures of Fe^3+^_3/2_ and Fe^3+1/2^, respectively (Appendix A). The C 1s (284.58 eV) and N 1s (397.58 eV) peaks in S1 correspond to the cyanide group connected to metal cations, while the two O 1s peaks at 529.68 eV and 531.78 eV indicate the presence of water molecules in the crystal structure [19]. For S2, the presence of Fe, Co, O, N, and C is also shown in Appendix A [20].

### Electrochemical Measurements for OER

PBA catalysts were used as an OER electrode in a standard three-electrode configuration in a 1.0 M KOH solution. The polarization curves of all catalysts are shown in Figure 5a. Prior to the OER reaction, the S3 PBA catalyst exhibited Ni oxidation peaks from 1.2–1.5 V, which can be attributed to the oxidation of Ni from lower oxidation states (Ni^0^, Ni^2+^) to higher ones (Ni^3+^ or Ni^4+^). The formation of these redox peaks suggests that the redox properties of Fe in S3 PBAs were tuned, and the synergistic interaction between Ni and Fe resulted in improved OER activity (Appendix A). Previous studies have shown that metal oxides and hydroxides with higher oxidation states are considered to be active catalysts for oxygen evolution processes [21,22,23]. The Ni/Fe synergistic interaction described in this study has been previously reported [23]. To evaluate the OER activity and Tafel slope, we used the LSV analysis. The PBA catalyst S3 manifested overpotential as low as 288 mV at 20 mA cm^−2^, which was substantially lower than S1 (364 mV) and S2 (340 mV). This indicates that PBAs (S3) had higher OER activity than PBs (S1), implying a Fe-Ni synergistic impact following Ni inclusion. The synergistic effect refers to the combined effect of materials (Ni/Fe) that is greater than the sum of their separate effects when used together (Ni or Fe) [24]. The addition of Ni in S3 catalyst results in more accessible active sites and an electron-rich environment, which improves OER performance. We also evaluated the Tafel slopes to verify the catalytic reaction kinetics of the catalysts (Figure 5b). A smaller Tafel slope value (86 mV dec^−1^) of S3 is obtained in comparison to S1 (93 mV dec^−1^) and S2 (90 mV dec^−1^). The low Tafel slope value indicates that the S3 catalyst favors fast OER kinetics.

Additionally, electrochemical impedance spectroscopy (EIS) analysis was conducted to examine the charge transfer kinetics of the electrocatalysts. As shown in Figure 5c, the Nyquist plots for S3 demonstrate the smallest semi-circle when compared to the S1 and S2 catalysts, indicating a superior charge transfer resistance (RCT) value with faster charge transfer and higher electronic conductivity, which confirms the excellent activity of the S3 catalyst. To comprehensively investigate the intrinsic activity, an analysis of the electrochemically active surface areas (ECSAs) was carried out using cyclic voltammetry (CV) tests at varying scan rates. This analysis determined the double layer capacitances (C_dl_), which directly indicate ECSAs. The results, presented in Appendix A, revealed C_dl_ values of 11.29 mF cm^−2^, 9.085 mF cm^−2^, and 8.27 mF cm^−2^ for samples S3, S2, and S1, respectively (Appendix A). These measured C_dl_ values offer valuable insights into both the extent of the surface area and the abundance of sites with electrocatalytic activity. Importantly, the C_dl_ value of S3 notably exceeded those of S2 and S1, indicative of a substantially larger surface area and a higher concentration of electrocatalytic active sites on the S3 sample. This outcome underscores the superior OER performance of the S3 sample compared to its counterparts. To thoroughly assess the catalysts’ intrinsic activity, we normalized steady-state polarization curves based on their electrochemically active surface areas (ECSAs), as seen in Appendix A. Impressively, S3 stood out by achieving a current density of 1.0 mA cm² at a notably lower overpotential of 440 mV, unlike S2 and S1. These results indicate that S3 has a significantly larger ECSA, showcasing the highest intrinsic electrocatalytic activity for the OER. The fact that S3 achieves this with a considerably reduced overpotential compared to S2 and S1 underscores its exceptional catalytic performance.

In addition to its impressive electrocatalytic activity, the catalyst’s electrochemical durability is crucial. Figure 5d and Appendix A illustrate a chronopotentiometry curve measured at both a fixed low current density of 20 mA cm^−2^ and a higher current density of 100 mA cm^−2^, assessing the stability of S3. Only slight overpotential changes were observed after 40 h at the lower current density and 24 h at the higher one, confirming the catalyst’s remarkable stability. The LSV curve of S3 provides further validation, plotted in the inset of Figure 5d after the 40 h chronopotentiometry test. This curve demonstrated minimal divergence from the initial data, affirming the electrocatalyst’s enduring performance. S3 exhibits excellent electrocatalytic activity and maintains its stability over prolonged operational periods, making it a promising OER catalyst for water splitting.

Taking into consideration all of the above results, we can conclude that the excellent OER performance of our S3 catalyst is due to the synergistic effects of the Fe and Ni species. Our hypothesis is that the presence of cyanide vacancies creates a local coordinatively unsaturated environment that modulates the oxidation states of both Ni and Fe in S3. Specifically, Fe sites in S3 are highly active for OER, while Ni ions serve as electrically conductive and stable hosts for the Fe locations. In S3, Fe atoms have a coordination number of six, which is reduced upon removal of CN groups, creating coordinatively unsaturated Fe sites. These active Fe sites can make bonds with oxygen (Fe-O bond) during OER.

It is noteworthy that both Ni and Fe in the S3 catalyst exist in the 2+ oxidation state. The partial metastable Fe^2+^ is readily oxidized to Fe^3+^ under ambient conditions. However, the presence of cyanide vacancies helps to restore the oxidation state of Fe^3+^, while the oxidation state of Ni^2+^ partially increases to 3+. This suggests that electron transfer occurs from the Ni ion to adjacent Fe sites, promoting the synergistic interaction between Ni and Fe [35,36,37,38]. Additionally, the presence of cyanide vacancies reduces the leaching of Fe species into the electrolyte, which promotes the formation of a Ni-Fe oxy(hydroxide) active site through a self-reconstruction of the PBA pre-catalyst during OER, further enhancing the OER activity of S3. This synthesis method is both simple and cost-effective and can produce gram-scale quantities of PB and PBAs (Appendix A). The decorating layer of PB proposes conduction avenues which hasten the electron transport and fundamentally enrich the electrochemical properties [39,40,41,42,43,44,45]. SEM of electrocatalyst S3 after OER is shown in Appendix A.

## 4. Conclusions

Our study presents, for the first time, a gram-scale mechanochemical grinding approach for synthesis of Prussian blue (PB) and its analog compounds from precursor salts. This cost-effective and straightforward synthetic method introduces carbon–nitrogen vacancies inherently into the double perovskite structure of PB and its analogs, which play a crucial role in their OER activity. Among the synthesized compounds, the S3 electrocatalyst exhibits excellent OER performance, thanks to its low charge resistance, high conductivity, and synergistic effect. Our fundamental research opens the way to developing a cost-effective and efficient electrocatalyst suitable for large-scale applications.

## Figures and Tables

**Figure 1 nanomaterials-13-02459-f001:**
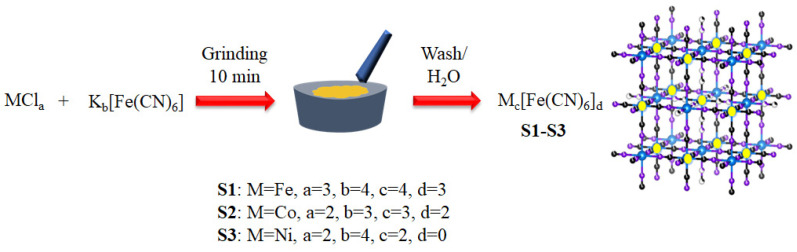
Schematic diagram for mechanochemical grinding synthesis of double perovskites like structures of Prussian blue and its analogous compounds.

**Figure 2 nanomaterials-13-02459-f002:**
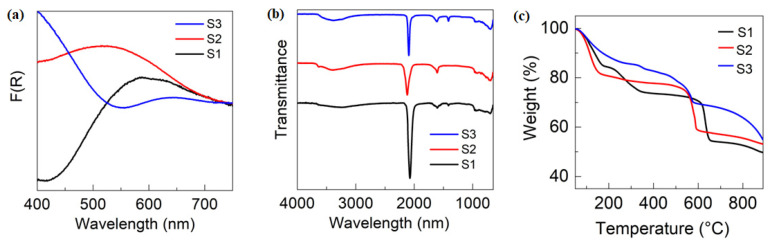
Characterization of S1, S2, and S3. (**a**) Absorption spectra, (**b**) FTIR spectra, and (**c**) thermogravimetric analysis (TGA).

**Figure 3 nanomaterials-13-02459-f003:**
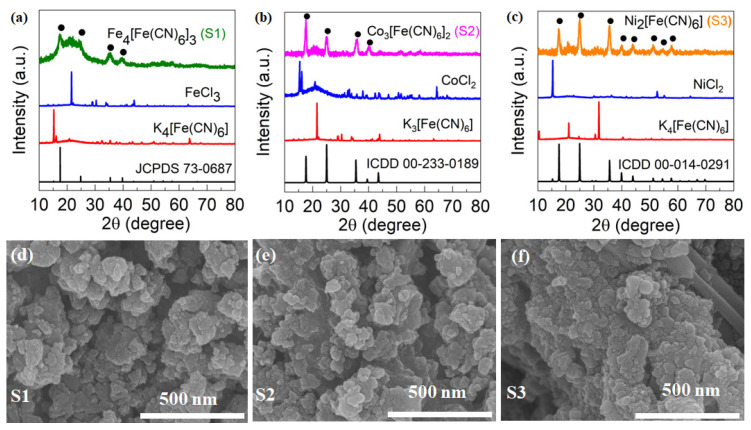
Physical characterizations. PXRD of (**a**) S1; (**b**) S2; (**c**) S3. SEM image of (**d**) S1; (**e**) S2; (**f**) S3.

**Figure 4 nanomaterials-13-02459-f004:**
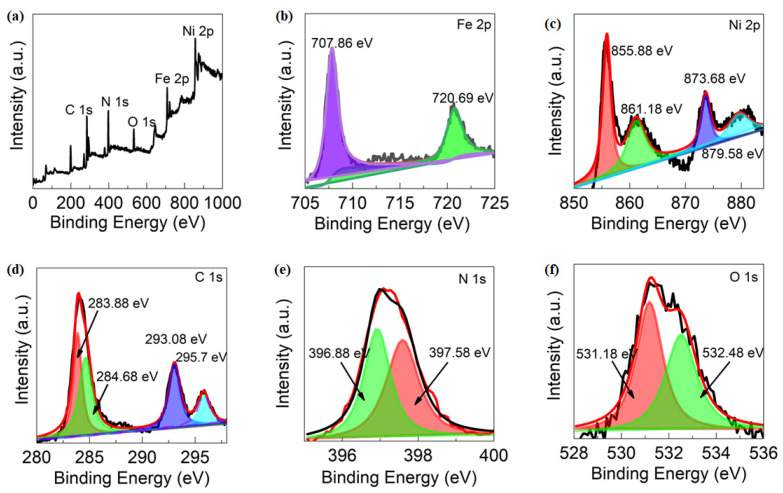
XPS spectra of S3. (**a**) Overall, XPS survey spectra. (**b**) Fe 2p. (**c**) Ni 2p. (**d**) C1s. (**e**) N1s. (**f**) O1s.

**Figure 5 nanomaterials-13-02459-f005:**
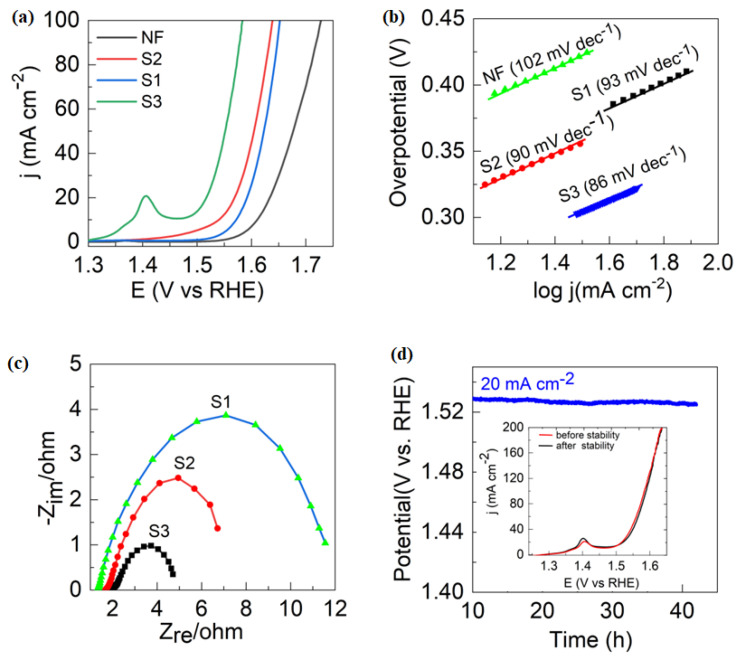
Electrocatalytic measurements of S3 for OER. (**a**) OER polarized curves for S1, S2, S3, and NF in 1 M KOH, (**b**) the corresponding Tafel plots for the OER, (**c**) Nyquist plots obtained, (**d**) chronopotentiometry curves of S3 at the current density of 20 cm^−2^; inset in (**d**) shows polarization curves of S3 before and after 40 h of stability.

## Data Availability

Data available upon request.

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
