# Peer review of "Microstructure and Oxygen Evolution Property of Prussian Blue Analogs Prepared by Mechanical Grinding"

_nanomaterials, 2023, doi:10.3390/nano13172459_

Round 1

Reviewer 1 Report

The authors prepared Fe4[Fe(CN)6]3, Co3[Fe(CN)6]2, Ni2[Fe(CN)6] catalyst through solvent-free mechanochemical synthetic strategy. The Ni analog manifests considerable electrocatalytic oxygen evolution reaction with overpotential of 288 mV to obtain the current density of 20 mA cm-2. The incorporating defectssuch and synergistic effects are attributed to high catalytic activity.

 I would suggest its acceptance for publication in Nanomaterials after a major revision. 

1.The stability test time needs to be extended and the stability needs to be tested at high current, such as 100 mA cm-2.

2.The catalyst electrochemical active area needs further study. The catalytic activity of a single catalytic site was studied by treating the catalytic properties with the electrochemical active area. 

3. For the integrity of the research background, more water-splitting literatures are suggested to be referred: Nanomaterials 2022, 12(7), 1227ï¼›International Journal of Hydrogen Energy, (2023), 48(33), 12176-12184; Layered double hydroxides as a robust catalyst for water oxidation through strong substrate-catalytic layer interaction, International Journal of Hydrogen Energy, https://doi.org/10.1016/j.ijhydene.2023.05.342  

4.The language and organization of the article need to be further strengthened to better highlight the innovation of the article.

5. Please explain why the starting point in Figure 5c is different?

6.The catalyst after stability should be characterized to explore the change of the catalyst active sites, the real active sites, and the reconstruction process?

7.Please check the note in Figure 3 carefully, it is inconsistent with the picture? S6?S7?

8.Figure S4 has an obvious error, only two lines, but shows four samples?

9.Figure S3 (e) O1s should be changed to (f) O1s. Figure S2 should be deleted (e) N.

10. Please check and correct the details of the article carefully.

The language and organization of the article need to be further strengthened to better highlight the innovation of the article. Please check and correct the details of the article carefully.

Reviewer 2 Report

Reviewer's comments:

In this work, the authors presented an innovative solvent-free mechanochemical synthetic strategy for synthesizing Fe4[Fe(CN)6]3, Co3[Fe(CN)6]2, Ni2[Fe(CN)6] with defects sites such as carbon-nitrogen vacancies introduced during the synthesis. Among all the synthesized PB analogs, the Ni analog manifested the considerable electrocatalytic oxygen evolution reaction (OER) with a low overpotential of 288 mV to obtain the current benchmark density of 20 mA cm-2. The incorporating defects, such as carbon-nitrogen vacancies, and synergistic effects were attributed to high catalytic activity of this material. This work paved the way for an easy and inexpensive large-scale production of earth-abundant non-toxic electrocatalysts with vacancy-mediated defects for OER. The authors well wrote this work and performed the experimental tests for OER and electrochemical tests. However, there are some issues to be well addressed before publish in Nanomaterials. 

Detailed comments:

1.  As mentioned by authors that defects play a promoting role in enhancing electrocatalytic performance of catalyst, some important works about the progress in defects of electrocatalysts should be cited here in the introduction part (Adv. Mater. 2020, 32, 1905923; Adv. Mater. 2017, 29, 1606459; Mater. Today Energy 2019, 12, 215).

2.  As shown in Figure 3a-c, the serial catalysts showed the clear peaks in XRD patterns, the authors can provide the TEM images of these samples in the manuscript.

3.  Electrochemical activity is closely related with surface area of catalysts. The authors should provide the data of electrochemical active surface areas of serial catalysts, which can provide the physicochemical property of materials (Angew. Chem. Int. Ed. 2022, 61, e202114899; Energy Environ. Sci. 2022, 15, 1201; ACS Catal. 2021, 11, 4498; ACS Catal. 2018, 8, 7585). Then, the LSV curves in Figure 5 can be normalized by according to the data of electrochemical active surface areas. In this case, we can easily compare the intrinsic catalytic activity of different catalysts. In addition, the above important references should be cited in this part.

4.  The characterization of S3 catalyst after OER test should be provided in the manuscript, which supports the excellent stability of its structure during OER, e.g., XRD, XPS etc.

Reviewer's comments:

In this work, the authors presented an innovative solvent-free mechanochemical synthetic strategy for synthesizing Fe4[Fe(CN)6]3, Co3[Fe(CN)6]2, Ni2[Fe(CN)6] with defects sites such as carbon-nitrogen vacancies introduced during the synthesis. Among all the synthesized PB analogs, the Ni analog manifested the considerable electrocatalytic oxygen evolution reaction (OER) with a low overpotential of 288 mV to obtain the current benchmark density of 20 mA cm-2. The incorporating defects, such as carbon-nitrogen vacancies, and synergistic effects were attributed to high catalytic activity of this material. This work paved the way for an easy and inexpensive large-scale production of earth-abundant non-toxic electrocatalysts with vacancy-mediated defects for OER. The authors well wrote this work and performed the experimental tests for OER and electrochemical tests. However, there are some issues to be well addressed before publish in Nanomaterials. 

Detailed comments:

1.  As mentioned by authors that defects play a promoting role in enhancing electrocatalytic performance of catalyst, some important works about the progress in defects of electrocatalysts should be cited here in the introduction part (Adv. Mater. 2020, 32, 1905923; Adv. Mater. 2017, 29, 1606459; Mater. Today Energy 2019, 12, 215).

2.  As shown in Figure 3a-c, the serial catalysts showed the clear peaks in XRD patterns, the authors can provide the TEM images of these samples in the manuscript.

3.  Electrochemical activity is closely related with surface area of catalysts. The authors should provide the data of electrochemical active surface areas of serial catalysts, which can provide the physicochemical property of materials (Angew. Chem. Int. Ed. 2022, 61, e202114899; Energy Environ. Sci. 2022, 15, 1201; ACS Catal. 2021, 11, 4498; ACS Catal. 2018, 8, 7585). Then, the LSV curves in Figure 5 can be normalized by according to the data of electrochemical active surface areas. In this case, we can easily compare the intrinsic catalytic activity of different catalysts. In addition, the above important references should be cited in this part.

4.  The characterization of S3 catalyst after OER test should be provided in the manuscript, which supports the excellent stability of its structure during OER, e.g., XRD, XPS etc.

Reviewer 3 Report

The paper is well conceived, the objectives are clearly presented in the introduction section. The preparation method is simple and therefore interesting for practical applications.

 The new materials obtained are analyzed using adequate means and techniques. FTIR analyses results are clear, TGA shows the water release, but in line 126 where some others peaks in TGA analyses are stated it is not clear why these state a possible use in high temperature catalysis ( I assume that below 500C no transformation occurs..?)

 XPS is very clearly explained and the figures ( Fig 4 S2 and S3)are relevant. Electrochemical measurements, mainly EIS, bring a good insight as the materials are aimed in OER catalysis.

The conclusions are relevant.

Round 2

Reviewer 1 Report

  • The article is ready to be received in its present form.